# QURL: EFFICIENT REINFORCEMENT LEARNING WITH QUANTIZED ROLLOUT

**Yuhang Li**[1,2]*, **Reena Elangovan**[2], **Xin Dong**[1], **Priyadarshini Panda**[2,3], **Brucek Khailany**[1]

[1] NVIDIA Research
[2] Yale University
[3] University of Southern California
yuhang.li@yale.edu, priya.panda@usc.edu
{relangovan, xdong, bkhailany}@nvidia.com

## ABSTRACT

Reinforcement learning with verifiable rewards (RLVR) has become a trending paradigm for training reasoning large language models (LLMs). However, due to the autoregressive decoding nature of LLMs, the rollout process becomes the efficiency bottleneck of RL training, consisting of up to 70% of the total training time. In this work, we propose Quantized Reinforcement Learning (QuRL) that uses a quantized actor for accelerating the rollout. We address two challenges in QuRL. First, we propose Adaptive Clipping Range (ACR) that dynamically adjusts the clipping ratio based on the policy ratio between the full-precision actor and the quantized actor, which is essential for mitigating long-term training collapse. Second, we identify the weight update problem, where weight changes between RL steps are extremely small, making it difficult for the quantization operation to capture them effectively. We mitigate this problem through the invariant scaling technique that reduces quantization noise and increases weight update. We evaluate our method with INT8 and FP8 quantization experiments on DeepScaleR and DAPO, and achieve 20% to 80% faster rollout during training.

## 1 INTRODUCTION

The emergence of reasoning Large Language Models (LLMs) such as OpenAI-O1 (Jaech et al., 2024) and DeepSeek-R1 (Guo et al., 2025) represents a fundamental transformation in AI capabilities through the strategic scaling of inference-time computation. By enabling extensive Chain-of-Thought (CoT) deliberation, these systems exhibit advanced problem-solving behaviors that deliver significant gains on challenging domains, notably mathematical reasoning (Luo et al., 2025b; Yu et al., 2025; Yue et al., 2025) and code synthesis (Luo et al., 2025a; Liu & Zhang, 2025). Such models sacrifice computational economy in favor of superior accuracy, producing elaborate reasoning chains that encompass systematic exploration, iterative verification, and strategic backtracking. Reinforcement learning (RL) forms the cornerstone of these breakthroughs. Through direct optimization on verifiable objective functions instead of proxy reward models, RL-driven approaches circumvent reward hacking pitfalls (Amodei et al., 2016; Wen et al., 2024) while maintaining stronger fidelity to genuine reasoning patterns.

A typical LLM RL training step comprises of three phases: actor rollout for response generation, forward pass to compute output probabilities, and backward pass for policy gradient updates. The autoregressive nature of LLMs imposes a fundamental bottleneck—each token must be decoded sequentially during rollout, requiring extensive memory bandwidth for weight and KV cache access. This sequential dependency severely limits parallelization opportunities. Consequently, the rollout phase dominates training time (He et al., 2025; Zheng et al., 2025), consuming approximately 70% of the total latency. This bottleneck is further exacerbated in reasoning tasks, where competitive performance requires extended CoT traces.

---

*Work done during internship at NVIDIA Research.

In this work, we propose Quantized Reinforcement Learning (QuRL), an efficient RL training algorithm through efficient inference. Specifically, we quantize the actor model for rollout while maintaining full-precision parameters for gradient updates. This approach transforms the on-policy RL into an off-policy setting: sequences are generated by a quantized actor while policy updates occur in the full-precision parameter space. This necessitates careful importance sampling and trust region constraints, as formalized in Decoupled PPO (Fu et al., 2025; Liu et al., 2025). However, we identify a critical failure mode where the decoupled PPO objective leads to training collapse at extended horizons, characterized by exponential growth of the divergence between the quantized actor and full precision actor. To address this instability, we propose Adaptive Clipping Range (ACR), which dynamically adjusts trust region bounds based on the policy divergence between the full-precision and the quantized actors.

Beyond importance sampling, we identify a critical scale mismatch between quantized and full-precision actors. RL updates usually satisfy trust region constraints (Schulman et al., 2015), resulting in weight changes that are orders of magnitude smaller than quantization errors. Consequently, the quantization operation fails to capture most weight updates, effectively decoupling the quantized model from the training dynamics. To address this fundamental mismatch, we propose Update-Aware Quantization (UAQ) that uses invariant scaling (Xiao et al., 2023a) to simultaneously reduce quantization error and amplify weight updates, ensuring that parameter changes exceed the quantization granularity threshold.

We validate our approach across multiple RL algorithms including PPO (Schulman et al., 2017), GRPO (Shao et al., 2024), and DAPO (Yu et al., 2025) on diverse reasoning benchmarks. Through 8-bit quantization (INT8 or FP8), we achieve substantial inference acceleration for 7B, 14B and 32B models, demonstrating 20%–80% throughput improvements. Our experimental evaluation demonstrates that QuRL consistently outperforms naive combinations of RL with quantized rollout as well as concurrent approaches (Liu et al., 2025). Notably, INT8 QuRL achieves 55.5% average accuracy on the DeepScaleR benchmark (Luo et al., 2025b) across five reasoning tasks, exceeding the baseline performance by 1.7%.

## 2 RELATED WORK

**Reinforcement Learning for Reasoning.** AI systems capable of extended reasoning constitute a distinct class of models that perform elaborate CoT deliberation prior to producing outputs, pioneered by OpenAI's o1 series (Jaech et al., 2024). Following this breakthrough, DeepSeek (Guo et al., 2025) and Kimi (Team et al., 2025) have documented comprehensive frameworks for developing reasoning models through RLVR. These contributions have established various RL algorithms as standard practice, including GRPO (Shao et al., 2024), Mirror Descent (Tomar et al., 2020), RLOO (Ahmadian et al., 2024), among others. However, this scaling comes at the cost of performing a significant amount of decoding, which severely under-utilizes modern hardware. To this end, the RL community has explored many ways for efficient training, including selective rollout generation (Zheng et al., 2025), rollout down-sampling (Xu et al., 2025), asynchronous multi-role distributed architectures (Fu et al., 2025). Despite these advances, achieving efficient RL training while maintaining model performance remains a key challenge.

**Quantization** Quantization has emerged as a fundamental technique for compressing and accelerating large-scale models. Comprehensive surveys by Gholami et al. (2022) and Nagel et al. (2021) provide systematic analyses of quantization advancements. This section reviews key quantization methods with emphasis on LLM applications. Quantization techniques fall into two main categories: Post-training Quantization (PTQ) and Quantization-Aware Training (QAT). PTQ methods operate directly on pre-trained models without additional training. Prominent approaches including Frantar et al. (2022b); Lin et al. (2023); Wei et al. (2022; 2023); Shao et al. (2023); Chee et al. (2023); Liu et al. (2023a) enhance uniform quantization through strategic optimization of weight parameters, scaling factors, and clipping boundaries. Alternative PTQ strategies explore non-uniform quantization schemes (Egiazarian et al., 2024; van Baalen et al., 2024; Elangovan et al., 2025) and mixed-precision architectures such as LLM.int8 (Dettmers et al., 2022). QAT methods integrate quantization into the training process itself. LLM-QAT (Liu et al., 2023b) addresses data requirements through synthetic generation, while Q-LoRA (Dettmers et al., 2023) combines quantization with low-rank adaptation to reduce memory overhead during fine-tuning.

## 3 PRELIMINARIES

We begin with a brief overview of the GRPO (Shao et al., 2024) algorithm. And then we introduce the basics of quantization operation and the subsequent challenges.

**Group Relative Policy Optimization.** GRPO adapts the PPO (Schulman et al., 2017) framework for training LLMs, notably by eliminating the need for a learned value function (critic). Instead of using generalized advantage estimation (GAE), GRPO estimates the advantage $\hat{A}_{i,t}$ at token $t$ of output $o_i$ based on the relative rewards within a group of $G$ outputs $\{o_1, o_2, \ldots, o_G\}$ sampled from the old policy $\pi_{\theta_{\mathrm{old}}}$ for the same prompt $q$. The objective function is:

$$\mathcal{J}_{\mathrm{GRPO}}(\theta) = \mathbb{E}_{q \sim P(q), o \sim \pi_{\theta_{\mathrm{old}}}} \left[ \frac{1}{G} \sum_{i=1}^{G} \frac{1}{|o_i|} \sum_{t=1}^{|o_i|} \min \left( R_{i,t} A_{i,t}, \mathrm{clip}(R_{i,t}, 1-\epsilon, 1+\epsilon) A_{i,t} \right) \right] \quad (1)$$

where $R_{i,t} = \frac{\pi_\theta(o_{i,t}|q_i)}{\pi_{\theta_{\mathrm{old}}}(o_{i,t}|q_i)}$ is importance sampling ratio between the old actor and the current actor on the $t$-th token of $i$-th generated response. Additionally, GRPO augments the PPO objective with an explicit KL regularization term $\mathcal{D}_{KL}(\pi_\theta||\pi_{\theta_{\mathrm{ref}}})$. The reference model $\pi_{\theta_{\mathrm{ref}}}$ (typically the initial supervised fine-tuned model) provides regularization, with the KL divergence computed using the k3 estimator from Schulman (2020).

**Quantization.** Quantization maps the full precision parameters $\theta$ into low-bit parameters $\hat{\theta} = Q(\theta)$. In general, a $b$-bit quantized parameter can be expressed as

$$Q(\theta, b) = \alpha \times (-1)^{\mathrm{sign}} \times 2^d \times (1 + \sum_{i=1}^{b-1-e} \frac{m_i}{2^i}), \quad (2)$$

where the representation consists of three components: sign, exponent, and mantissa, scaled by a factor $\alpha$. Here, $\mathrm{sign} \in \{-1, +1\}$ encodes the sign, $d \in [1, 2^e]$ represents the exponent using $e$ bits. And the mantissa uses the remaining $(b - 1 - e)$ bits with $m_i \in \{0, 1\}$. When $e = 0$, this formulation reduces to integer quantization. The scaling factor $\alpha$ is determined by the maximum absolute value within a group of weights or activations. The granularity of quantization depends on the group size, ranging from channel-wise to block-wise operations. To further reduce memory overhead, the scaling factor itself can be quantized, as in NVFP4 (NVIDIA, 2025).

## 4 QURL: QUANTIZED REINFORCEMENT LEARNING

To accelerate the rollout phase, we quantize the weights and activations of the old actor model $\theta_{\mathrm{old}}$ to lower-bit representations, enabling efficient matrix multiplication during inference $\hat{\theta}_{\mathrm{old}} = Q(\theta_{\mathrm{old}}, b)$. Fig. 1 illustrates the pipeline for incorporating quantization into the RL.

Our QuRL approach occupies a unique position between post-training quantization (PTQ) and quantization-aware training (QAT). Unlike QAT, we do not explicitly optimize quantization performance through gradient descent—the actor undergoes one-shot quantization before deployment for rollout. However, unlike pure PTQ, the actor parameters are implicitly influenced by the gradients computed from the quantized model's outputs during policy updates. This dual nature imposes specific re-

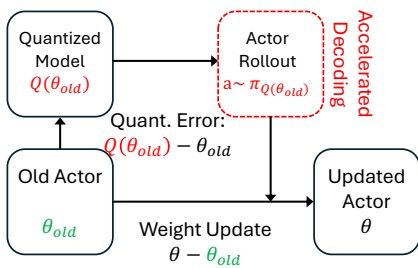

Figure 1: Overview of QuRL training. The sampling model $\theta_{\mathrm{old}}$ is quantized to $\hat{\theta}_{\mathrm{old}}$ for rollout.

quirements on our quantization strategy: it must be sufficiently simple to avoid complex calibration procedures while remaining expressive enough to preserve the learning dynamics. In the following sections, we present our methodology addressing both the reinforcement learning adaptations and the quantization operations necessary for efficient training.

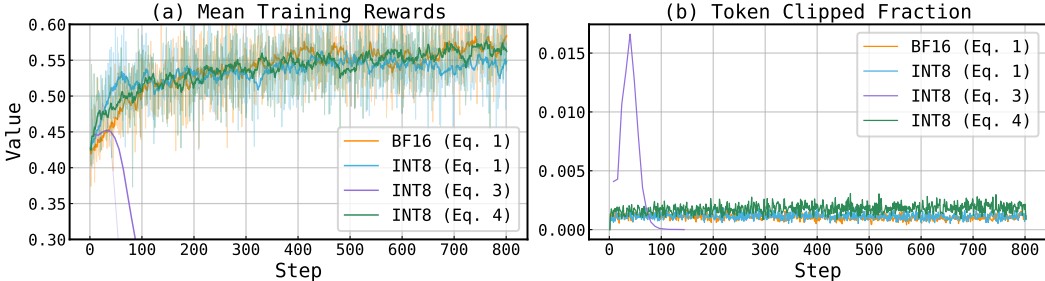

Figure 2: Comparison of (a) training rewards and (b) token clipped fraction under different training objective or quantization.

### 4.1 ISSUES WITH IMPORTANCE SAMPLING AND CLIPPING

Given that rollout data has been sampled from the quantized actor, we can rewrite the RL objective as

$$
\mathcal{J}(\theta) = \mathbb{E}_{q \sim P(q), o \sim \pi_{\hat{\theta}_{\mathrm{old}}}} \left[ \frac{1}{G} \sum_{i=1}^{G} \frac{1}{|o_i|} \sum_{t=1}^{|o_i|} \min \left( \hat{R}_{i,t} A_{i,t}, \mathrm{clip}(\hat{R}_{i,t}, 1 - \epsilon, 1 + \epsilon) A_{i,t} \right) \right], \quad (3)
$$

where $\hat{R}_{i,t} = \frac{\pi_\theta(o_{i,t}|q_i)}{\pi_{\hat{\theta}_{\mathrm{old}}}(o_{i,t}|q_i)}$ denotes the importance sampling ratio between the current full-precision actor $\pi_\theta$ and the quantized old actor $\pi_{\hat{\theta}_{\mathrm{old}}}$.

We implement the objective of Eq. (3) and compare it with full-precision GRPO experiments. The model and dataset follow the DeepScaleR setup (Luo et al., 2025b). Unfortunately, the quantized model's importance sampling leads to training instability, with rewards collapsing after several RL steps as shown in Fig. 2. Analysis of the token clipped fraction in Fig. 2(b) reveals that $\hat{R}_{i,t}$ exhibits significantly higher clipping rates than the full-precision baseline. The fraction rapidly increases to 1.5% before abruptly dropping to zero, indicating severe instability in the $\hat{R}_{i,t}$ when applying clipping on top of it. Additionally, we test the objective of Eq. (1) with quantized rollout, which instead uses the full-precision old actor as the denominator for clipping and importance sampling. Fig. 2 shows that this objective can have a stable training curve, but may produce a large gap between BF16 after 800 steps of RL training.

To address the instability in Eq. (3), we adopt the decoupled PPO objective (Hilton et al., 2022; Fu et al., 2025), which separates the *behavior policy* $\pi_{\theta_{\mathrm{behav}}}$ (for token sampling) from the *proximal policy* $\pi_{\theta_{\mathrm{prox}}}$ (for clipping):

$$
\mathcal{J}_{\mathrm{decoupled}}(\theta) = \tilde{\mathbb{E}}_{o \sim \pi_{\theta_{\mathrm{behav}}}} \left[ \frac{\pi_{\theta_{\mathrm{prox}}}(o_{i,t})}{\pi_{\theta_{\mathrm{behav}}}(o_{i,t})} \min \left( R_{i,t} A_{i,t}, \mathrm{clip}(R_{i,t}, 1 - \epsilon, 1 + \epsilon) A_{i,t} \right) \right], \quad (4)
$$

where $R_{i,t} = \frac{\pi_\theta(o_{i,t})}{\pi_{\theta_{\mathrm{prox}}}(o_{i,t})}$ denotes the ratio between the current policy and the proximal policy. For simpler notation, we integrate the averaging across responses and groups into expectation $\tilde{\mathbb{E}}$, as they do not change the clipping/importance sampling outcome. **In QuRL, we set the behavior policy as the quantized old actor ($\pi_{\theta_{\mathrm{behav}}} = \pi_{\hat{\theta}_{\mathrm{old}}}$) and the proximal policy as the full-precision old actor ($\pi_{\theta_{\mathrm{prox}}} = \pi_{\theta_{\mathrm{old}}}$).** Compared to $\hat{R}_{i,t}$ that uses a quantized actor to determine clipping, $R_{i,t}$ enables more tokens to be trained via correct importance sampling. As shown in Fig. 2, this approach significantly improves training stability.

FlashRL (Liu et al., 2025) observes that $\pi_{\hat{\theta}_{\mathrm{old}}}$ is usually obtained from the inference engine such as vLLM (Kwon et al., 2023) and SGLang (Zheng et al., 2024). However, due to the implementation difference between training (i.e., HuggingFace and Megatron) and inference (i.e., vLLM and SGLang) engines, an extra engineering discrepency between $\pi_{\theta_{\mathrm{prox}}}$ and $\pi_{\theta_{\mathrm{behav}}}$ is introduced and will hinder RL training. FlashRL proposes Truncated Importance Sampling (TIS) to reduce this

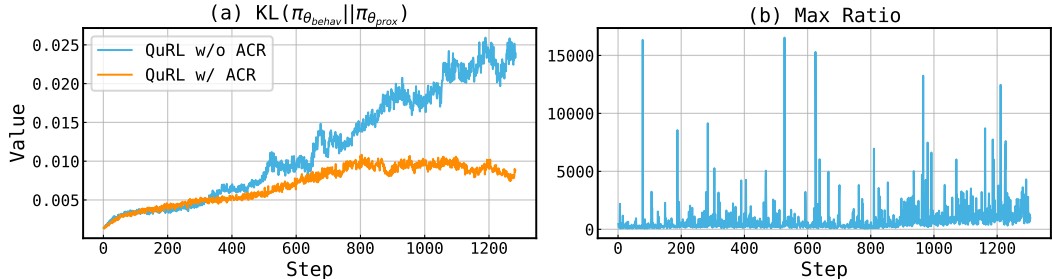

Figure 3: Training dynamics of QuRL. (a) Training collapses after 1000 steps due to increased KL divergence between behavior and proximal policy, and (b) the maximum value of the proximal-to-behavior policy ratio.

difference, given by

$$\mathcal{J}_{\text{TIS}}(\theta) = \tilde{\mathbb{E}}_{o \sim \pi_{\theta_{\text{behav}}}} \left[ \min \left( \frac{\pi_{\theta_{\text{prox}}}(o_{i,t})}{\pi_{\theta_{\text{behav}}}(o_{i,t})}, C \right) \min \left( R_{i,t} A_{i,t}, \text{clip}(R_{i,t}, 1 - \epsilon, 1 + \epsilon) A_{i,t} \right) \right]. \tag{5}$$

where $C$ bounds the proximal-to-behavior ratio. This formulation reduces computational overhead by directly accessing probabilities from the inference engine and naturally extends to other off-policy RL methods. However, even with these modifications, the decoupled objective alone cannot fully bridge the quantization gap in QuRL, particularly during later training stages.

## 4.2 ADAPTIVE CLIPPING RANGE

The TIS essentially modifies the behavior policy by truncating the behavior policy if it is extremely small. To see this, we define

$$\pi_{\theta_{\text{behav}}^{\text{trunc}}} = \max(\pi_{\theta_{\text{behav}}}, \frac{\pi_{\theta_{\text{prox}}}}{C}). \tag{6}$$

Now, we can rewrite TIS objective as the decoupled PPO objective form, given by

$$\mathcal{J}_{\text{decoupled}}(\theta) = \tilde{\mathbb{E}}_{o \sim \pi_{\theta_{\text{behav}}}} \left[ \frac{\pi_{\theta_{\text{prox}}}(o_{i,t})}{\pi_{\theta_{\text{behav}}^{\text{trunc}}}(o_{i,t})} \min \left( R_{i,t} A_{i,t}, \text{clip}(R_{i,t}, 1 - \epsilon, 1 + \epsilon) A_{i,t} \right) \right], \tag{7}$$

Essentially, the gradient of the original decoupled PPO objective is scaled by $r_{i,t} = \pi_{\theta_{\text{behav}}}(o_{i,t}) / \pi_{\theta_{\text{behav}}^{\text{trunc}}}(o_{i,t})$. As shown in Fig. 3(b), the maximum proximal-to-behavior ratio can reach up to $10^5$, causing an extremely large gradient norm if using the decoupled PPO objective. The above equation effectively avoids the excessive gradient norm.

In practice, we find that TIS works well under 500 steps of RL training. However, at long training steps (e.g., $> 1000$ steps), we observe the KL divergence between the behavior policy and the proximal policy (i.e., $\mathcal{D}_{KL}(\pi_{\theta_{\text{behav}}} || \pi_{\theta_{\text{prox}}}) = \mathbb{E}[\log \frac{\pi_{\theta_{\text{behav}}}}{\pi_{\theta_{\text{prox}}}}]$) continues to increase. As shown in Fig. 3(a), the KL divergence increases from 0.002 to 0.025, which is $12\times$ higher. This indicates that the truncated behavior policy also leads to biased gradient estimation, especially for large $r$.

To mitigate this problem, we examine the clipping mechanism in the decoupled PPO objective and propose the Adaptive Clipping Range (ACR). Our intuition is that, when the behavior policy is truncated, the factor $r_{i,t}$ implicitly affects the clipping. More concretely, given that $0 < r_{i,t} \leq 1$, we can absorb this factor into the clipping term as well as its range, given by

$$r_{i,t} \text{clip}(R_{i,t}, (1 - \epsilon), (1 + \epsilon)) = \text{clip}(r_{i,t} R_{i,t}, r_{i,t}(1 - \epsilon), r_{i,t}(1 + \epsilon)). \tag{8}$$

This operation shrinks both the upper and lower clipping range by a factor of $r_{i,t}$. For negative advantage sequences, it does affect the clipping since the large ratios do not get clipped regardless. However, for positive advantage sequences, it reduces the upper bound. For large $r_{i,t}$ where the difference is likely due to training/inference engine execution, its biased estimation unexpectedly clips more tokens. To address this issue, we propose to use a fixed upper threshold $(1 + \epsilon)$, to allow

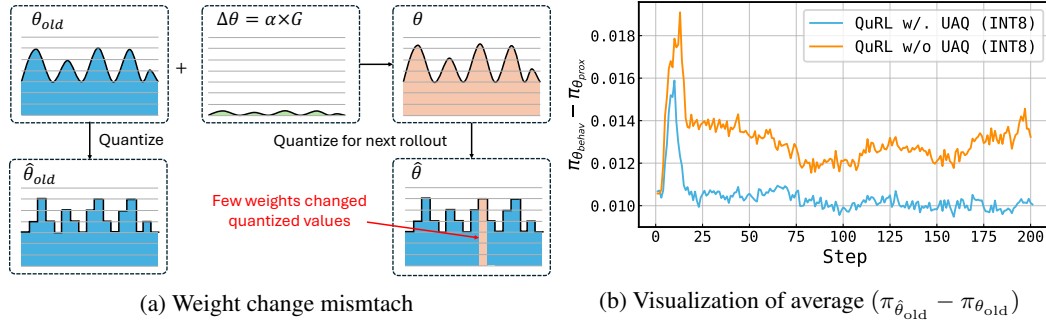

(a) Weight change mismtach                (b) Visualization of average $(\pi_{\hat{\theta}_{\text{old}}} - \pi_{\theta_{\text{old}}})$

Figure 4: The weight update problem. (a) An intuitive example showing weight quantization cannot sense the weight update suitably and (b) visualization of training dynamics showing the difference between $\pi_{\hat{\theta}_{\text{old}}}$ and $\pi_{\theta_{\text{old}}}$.

more tokens to pass if the $\pi_{\theta_{\text{behav}}}(o_{i,t})$ is truncated. As a result, we can rewrite our ACR into:

$$\mathcal{J}_{\text{ACR}}(\theta) = \tilde{\mathbb{E}}_{o \sim \pi_{\theta_{\text{behav}}}} \left[ \min \left( \frac{\pi_{\theta_{\text{prox}}}(o_{i,t})}{\pi_{\theta_{\text{behav}}}(o_{i,t})}, C \right) \min \left( R_{i,t} A_{i,t}, \text{clip} \left( R_{i,t}, (1-\epsilon), \frac{(1+\epsilon)}{r_{i,t}} \right) A_{i,t} \right) \right].$$
(9)

The ACR can dynamically adjust the clipping range based on the proximal-to-behavior ratio: For tokens where $\frac{\pi_{\theta_{\text{prox}}}(o_{i,t})}{\pi_{\theta_{\text{behav}}}(o_{i,t})} > C$, $r_{i,t} < 1$ enlarges the upper clipping bound, allowing more positive tokens to be updated. Otherwise, the threshold is the same as TIS.

### 4.3 UPDATE-AWARE QUANTIZATION

Another crucial challenge in QuRL is the mismatch of magnitude between the weight quantization change and the weight update change. During each RL step, $\theta_{\text{old}}$ is quantized to $\hat{\theta}_{\text{old}}$ for rollout, then updated to $\theta$ to become the old actor for the next step. The weight change magnitudes follow:

$$\hat{\theta}_{\text{old}} - \theta_{\text{old}} \propto \frac{|\theta_{\text{old}}|}{2^b}, \qquad \theta - \theta_{\text{old}} \propto \alpha G,$$
(10)

where $\alpha$ denotes the learning rate and $G$ the gradient. In typical RL experiments, we observe $G \in [0.1, 1.0]$ with $\alpha = 10^{-6}$, yielding weight updates of order $10^{-7}$ to $10^{-6}$. This is substantially smaller than the quantization error, as the weight norm itself ranges from $(0.001, 0.1)$. We also provide an example in Fig. 4(a) to illustrate this problem.

Empirical analysis confirms this mismatch. Comparing $\hat{\theta}_{\text{old}}$ across RL steps with INT8 quantization in DeepScaleR experiments, the update is much smaller than quantization error. See Appendix A for more details. This indicates that quantization masks nearly all weight updates, effectively freezing the quantized model despite ongoing training. In Fig. 4(b), we measure the average difference between $\pi_{\hat{\theta}_{\text{old}}}$ and $\pi_{\theta_{\text{old}}}$ of INT8 quantization in DAPO task (Yu et al., 2025).

Since QuRL operates between PTQ and QAT paradigms, neither approach offers an optimal solution. Complex calibration algorithms like GPTQ (Frantar et al., 2022a) could theoretically capture finer weight changes if applied at each step, but would impose prohibitive training time overhead on rollout. For QAT, it will introduce additional discrepancies between training and inference engines, exacerbating importance sampling bias (Liu et al., 2025).

To mitigate this problem, we propose Update-Aware Quantization (UAQ), a one-time weight adjustment performed before RL training begins. Our approach leverages *invariant scaling* of linear layers in transformer blocks (Xiao et al., 2023b). Given weights $W$ and input activations $X$ in a layer, invariant scaling preserves the output by

$$WX = \left( \frac{W}{s} \right) \cdot (sX).$$
(11)

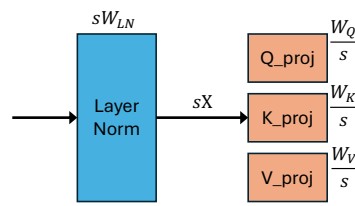

Figure 5: Invariant scaling of Q/K/V layers.

The scale $s$ is applied column-wise to $W$ and row-wise to $X$.
The activation scaling can be absorbed into the preceding layer
(e.g., LayerNorm), as illustrated in Fig. 5.

Unlike existing PTQ methods (Xiao et al., 2023b) that minimize quantization error $\|Q(W/s)Q(sX) - WX\|$, we strategically choose $s > 1$ to balance weight quantization error against update magnitude such that

$$\hat{\theta}_{\text{old}} - \theta_{\text{old}} \propto \frac{|\theta_{\text{old}}|}{s \cdot 2^b}, \quad \theta - \theta_{\text{old}} \propto s \cdot \alpha G. \tag{12}$$

The scaling factor $s$ reduces quantization error by a factor of $s$ while amplifying weight updates by the same factor. The weight update amplification occurs because gradients with respect to $W$ are computed as $\nabla_W \mathcal{L} = (\nabla_Y \mathcal{L}) X^\top$, where the pre-scaled activations $X$ have been multiplied by $s$.

This dual effect creates an $s^2$ improvement in the ratio between weight updates and quantization noise, enabling the quantized model to capture training dynamics more effectively. Empirically, we find $s = 1.5$ provides consistent improvements on INT8 and FP8 quantization, striking an effective balance between reducing quantization artifacts and maintaining numerical stability.

## 5 EXPERIMENTS

All our experiments were conducted with the hybrid engine-based RL framework, VeRL (Sheng et al., 2024). We evaluate QuRL across three distinct reinforcement learning configurations: (1) PPO training on GSM8K (Cobbe et al., 2021), (2) DAPO (Yu et al., 2025) optimization on AIME mathematical reasoning tasks, and (3) GRPO training on the DeepScaleR benchmark (Luo et al., 2025b). Our quantization experiments employ both INT8 and FP8 precision formats. Weight quantization utilizes channel-wise scaling factors, while activation quantization applies token-wise scaling. We leverage vLLM's optimized INT8 and FP8 matrix multiplication kernels (Kwon et al., 2023) to achieve computational acceleration during inference. Note that FP8 KV cache quantization remains suboptimally implemented in the current vLLM version and does not yield measurable throughput improvements; consequently, we exclude KV cache quantization from our experimental evaluation.

### 5.1 REASONING RESULTS

**PPO on GSM8K.** We evaluate the PPO algorithm (Schulman et al., 2017) on the GSM8K dataset, which comprises 7.4k training examples and 1.3k validation examples. We use Qwen2.5-0.5B-Instruct (Qwen et al., 2024) as the base model and conduct training over 15 epochs, corresponding to 435 RL optimization steps. Training employs a batch size of 256 with a maximum response length of 512 tokens per rollout. Evaluation metrics are computed using greedy decoding on the test set (deterministic next-token selection based on maximum probability).

We compare against four experimental configurations: (1) Full-precision RL baseline, (2) Quantized rollout with standard importance sampling—applying Equation 1 to responses sampled from the quantized policy $o \sim \pi_{\hat{\theta}_{\text{old}}}$, (3) FlashRL with Truncated Importance Sampling (TIS) (Liu et al., 2025), and (4) QuRL with Adaptive Clipping Range. Note that Update-Aware Quantization is disabled for this experiment due to the relatively high learning rate ($10^{-5}$), which already provides sufficient weight update magnitude. Table 1 presents final checkpoint accuracies. The results demonstrate that naive INT8 quantization without decoupled behavior/proximal policies yields substantial performance degradation. FlashRL (Liu et al., 2025) achieves training stabilization through TIS and decoupled PPO, yet maintains a notable accuracy gap relative to BF16 baseline—particularly severe under INT8 quantization (4% degradation). In contrast, QuRL with ACR reduces this gap to 2% for INT8 and approximately 1% for FP8, demonstrating superior quantization robustness across precision formats.

**DAPO on AIME 2024.** Next, we test the decoupled clip and dynamic sampling policy optimization (DAPO) (Yu et al., 2025) with Qwen2.5-7B-Math. We use the 17k dataset from the original paper and apply the decoupled clip where $\epsilon_{\text{high}} = 0.28$ and $\epsilon_{\text{low}} = 0.2$ for the default clipping range. We optimize the base model for 200 steps, and the learning rate is set to $1e - 6$. In each step, we sample 512 queries and 16 rollout responses per query. Additionally, DAPO does not apply any KL divergence loss between the actor and reference models. For evaluation, we use two metrics (Avg@1)

Table 1: Comparison of GSM8k accuracy.

| Method | Bitwidth | Accuracy |
|---|---|---|
| RL | BF16 | 55.35 |
| RL | INT8 | 48.78 |
| FlashRL (Liu et al., 2025) | INT8 | 51.40 |
| QuRL (Ours) | INT8 | 53.55 |
| RL | FP8 | 0.0 |
| FlashRL (Liu et al., 2025) | FP8 | 53.60 |
| QuRL (Ours) | FP8 | 54.28 |

Figure 6: Convergence of INT8 experiment.

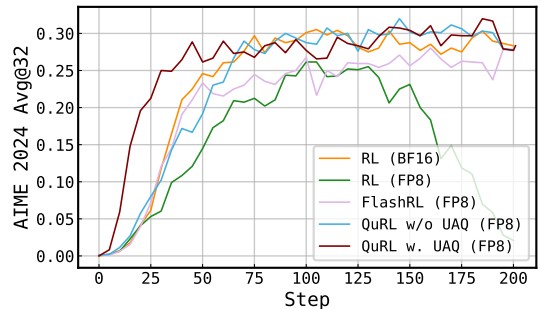

Table 2: Comparison of AIME 2024 accuracy.

| Method | Bitwidth | Avg@1 | Avg@32 |
|---|---|---|---|
| RL | BF16 | 33.33 | 31.67 |
| RL | INT8 | 0.00 | 0.001 |
| FlashRL | INT8 | 26.66 | 30.29 |
| QuRL w/o UAQ | INT8 | 33.33 | 30.63 |
| QuRL w/ UAQ | INT8 | 33.33 | 31.25 |
| RL | FP8 | 0.00 | 0.003 |
| FlashRL | FP8 | 30.00 | 32.60 |
| QuRL w/o UAQ | FP8 | 36.66 | 33.12 |
| QuRL w/ UAQ | FP8 | 33.33 | 33.27 |

Figure 7: Convergence of FP8 experiment.

and (Avg@32) on the AIME 2024 dataset, where, Avg@1 represents the accuracy achieved using greedy decoding (deterministic next token prediction) and Avg@32 represents the average accuracy of 32 sampled responses per problem, using a temperature of 1.0 and a top $p$ of 0.7.

As shown in Table 2, vanilla INT8/FP8 RL has near 0 accuracy on the AIME 2024 dataset, indicating that a biased estimation of the importance sampling will result in crashed performance. The convergence figure on the right shows that RL can converge well in the first 100 steps, but results in decreased performance for the latter 100 steps. This is due to the increased gap between proximal and behavioral policy through training. FlashRL (Liu et al., 2025) converges better than RL and has much better final accuracy than RL. For example, with INT8 quantization, FlashRL achieves 30.3% Avg@32 accuracy, with a 1.4% gap from the full precision baseline. Our QuRL, equipped with ACR, can successfully close the gap during training. The final accuracy also shows improved results, with 33.1% Avg@32 accuracy under FP8 quantization.

**GRPO on DeepScaleR.** Finally, we test the performance of our algorithm on an open-source project, DeepScaleR (Luo et al., 2025b), which improves the reasoning boundaries of DeepSeek-Distill-Qwen1.5B models (Guo et al., 2025). The training dataset contains 40k math problems from AIME problems from 1983 to 2023, as well as Omni-Math Gao et al. (2024) and Still (Min et al., 2024). We train the actor for 3 stages, under 8k, 16k, and 24k context length, respectively. The training batch size is 256, with the learning rate of $1e - 6$. Following the official implementation,

Table 3: Comparison of Avg@32 accuracy across various math reasoning tasks of DeepScaleR.

| Method | Bitwidth | AIME24 | AMC | MATH | Minerva | Olympiad | Avg |
|---|---|---|---|---|---|---|---|
| Base | BF16 | 28.54 | 62.58 | 82.90 | 26.38 | 43.58 | 48.80 |
| RL | BF16 | 40.73 | 73.45 | 87.71 | 30.56 | 49.59 | 56.40 |
| RL | INT8 | 33.95 | 68.75 | 84.90 | 28.12 | 45.85 | 52.31 |
| FlashRL | INT8 | 36.77 | 70.55 | 85.88 | 28.44 | 47.33 | 53.80 |
| QuRL w/o UAQ | INT8 | 39.06 | 70.48 | 86.48 | 29.20 | 48.75 | 54.79 |
| QuRL w/ UAQ | INT8 | 40.52 | 71.34 | 87.20 | 29.22 | 49.13 | 55.48 |

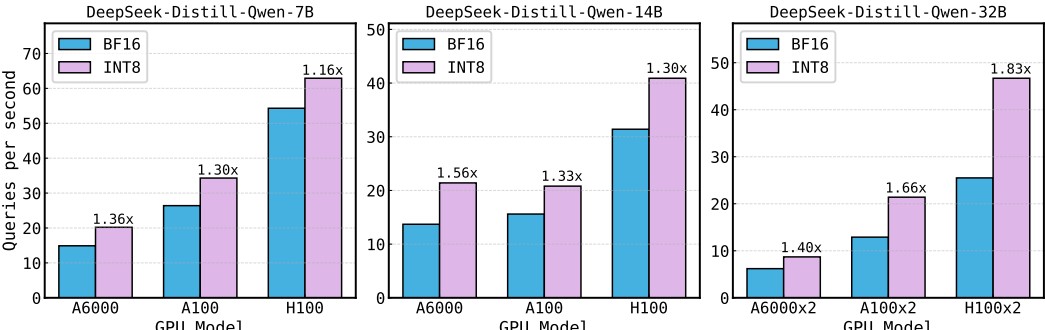

Figure 8: Inference acceleration of INT8 quantization.

for the first stage, we generate 8 rollouts per query with 8k context length and train the model for 800 steps. For the latter two stages, 16 rollouts per query are generated, and the actor is trained for 400 steps. The coefficient for KL divergence in GRPO is set to $1e - 3$. The temperature is set to 0.6

For evaluation, we follow the official implementation to compare the Avg@32 results on AIME 2024 (Li et al., 2024), MATH 500 (Hendrycks et al., 2021), AMC 2023 (Ouyang et al., 2022), Minerva Math (Lewkowycz et al., 2022), and Olympiad Bench (He et al., 2024) as well as the average accuracy of all above. The results are shown in Table 3. The full-precision RL improves the base model by 7.6% average accuracy across 5 tasks and notably 12% accuracy improvement on the AIME 2024 dataset. Instead, INT8 RL can barely improve the base model accuracy, for example, 5% improvement on the AIME 2024 dataset. Averaging all tasks, INT8 RL has a large gap of 4.1% compared to BF16 RL. FlashRL achieves a slight improvement over INT8 RL, with a 1.5% higher average accuracy among 5 tasks. On the other hand, QuRL w/ UAQ significantly boosts the average accuracy of INT8 RL by 3%.

## 5.2 THROUGHPUT TEST

In this section, we demonstrate the throughput benefits of applying quantization during rollout (decoding). We use the script from GuideLLm (Neural Magic, 2024) to evaluate the DeepSeek-Distill-Qwen-7B/14B/32B models on the vLLM platform (Kwon et al., 2023). We test INT8 quantization across multiple GPU types, including A6000, A100, and H100. For 7B and 14B model, we evaluate the throughput (queries per second) on one GPU, while for 32B model, we evaluate it with tensor parallelism across 2 GPUs.

The results are shown in Fig. 8. For the 7B model, INT8 quantization can bring 20%~30% acceleration effect, while for the 32B model, we find INT8 quantization can bring 70% faster throughput on A100 and 90% faster throughput on H100. Generally, we observe that larger models benefit more from quantization. This is due to the large model being bottlenecked by matrix multiplication, yet smaller models are usually bottlenecked by the I/O Nevertheless, we emphasize that QuRL is compatible with other types of compression (Frantar & Alistarh, 2023; Liu et al., 2024).

## 5.3 ABLATION STUDY

In this section, we compare the results of choosing different scales for UAQ. On one hand, a large scale contributes to a smaller gap between weight update and weight quantization noise. On the other hand, it will also make the weight update more than usual, causing more clipped tokens and decreasing the RL performance. To demonstrate this effect, we compare scales $s = 1, 1.5, 2$ and also test another alternative by directly increasing the learning rate on the DAPO tasks with INT8 quantization. The results are shown in Table 4. It can be seen that $s = 1.5$ with the original learning rate provides the best results. The larger scale or learning rate results in less stable training of RL and decreases the accuracy on AIME 2024.

| Scale $s$ | Learning Rate | Avg@32 |
|---|---|---|
| $s = 1$ | $\alpha = 10^{-6}$ | 30.63 |
| $s = 1.5$ | $\alpha = 10^{-6}$ | 31.25 |
| $s = 2$ | $\alpha = 10^{-6}$ | 29.15 |
| $s = 1$ | $\alpha = 1.5 \times 10^{-6}$ | 29.06 |
| $s = 1$ | $\alpha = 2 \times 10^{-6}$ | 26.66 |

Table 4: Ablation on scale and $\alpha$.

## CONCLUSION

We present QuRL, an efficient RL training method that accelerates rollout generation through quantized inference. QuRL addresses two fundamental challenges: clipping instability and weight update-quantization mismatch. Our ACR prevents training collapse by dynamically adjusting the clipping, while UAQ bridges the scale gap between weight updates and quantization errors through invariant scaling. Experiments across multiple RL algorithms demonstrate consistent improvements over baselines and 20~80% throughput improvement over BF16 rollout.

## ACKNOWLEDGMENT

The authors would like to thank Charbel Sakr for his helpful feedback. This work was supported in part by the National Science Foundation (CAREER Award, Grant #2312366, Grant #2318152), the DARPA Young Faculty Award, the DoE MMICC center SEA-CROGS (Award #DE-SC0023198) and the Global Industrial Technology Cooperation Center (GITCC) program.

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

## A  WEIGHT CHANGE VISUALIZATION

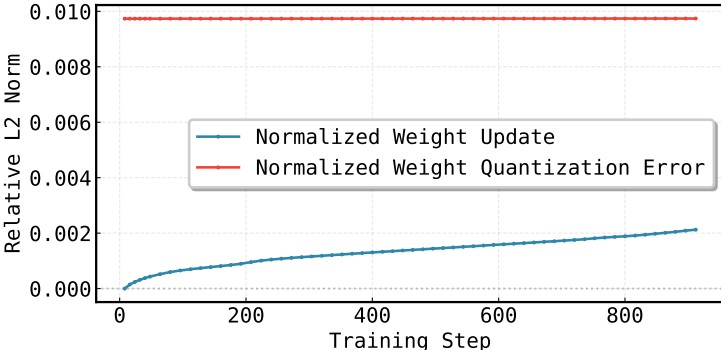

Figure 9: Visualization of normalized weight changes.

In this section, we visualize the weight change during RL learning. We conduct our analysis on the DeepScaleR task (Luo et al., 2025b). Our metric include (1) Normalized Weight Update, defined as

$$\text{NormalizedWeightUpdate}(t) = \frac{||\theta^{t+1} - \theta^t||_F^2}{||\theta^t||_F^2}, \tag{13}$$

where $\theta^t$ denotes the actor weights after $t$ RL steps. This metric captures the ratio between the total weight update and the initial weight values. The second metric we adopt is the Normalized Weight Quantization Error, defined as

$$\text{NormalizedWeightQuantError} = \frac{||Q(\theta^t) - \theta^t||_F^2}{||\theta^t||_F^2}. \tag{14}$$

We use INT8 quantization, and compare their values in Fig. 9. It can be found that the weight quantization error is much larger than the weight update, especially at the early training stages. Note that the update are measured across every 16 steps, which means the actual weight update per step could be far smaller than the plotted values.

## B  DEEPSCALER VISUALIZATION

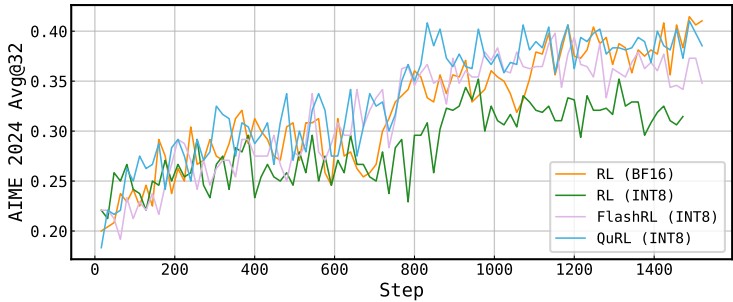

Figure 10: Visualization of test accuracy (AIME 2024 Avg@32).

Here, we provide the test accuracy of our DeepScaleR experiments. It can be observed that in the long-horizon training scenarios, INT8 RL incurs a large gap with the BF16 RL. Although FlashRL ensures consistent improvement before 1200 steps, its test accuracy starts to drop after 1200 steps. While QuRL can have consistent improvement across the whole training cycle.

