# OpenReview forum: "QuRL: Low-Precision Reinforcement Learning for Efficient Reasoning"
_ICLR.cc/2026/Conference — ICLR 2026 Poster_

### Official Review · Reviewer_xNJr · 2025-10-31

**Soundness:** 3
**Presentation:** 3
**Contribution:** 3
**Rating:** 6
**Confidence:** 3

**Summary:**

This paper introduces QuRL (Quantized Reinforcement Learning), a novel method to accelerate the rollout phase in reinforcement learning (RL) for large language models (LLMs). The authors address the critical bottleneck in RL training—where the autoregressive decoding process consumes most of thel training time—by quantizing the actor model during rollout while maintaining full-precision parameters for policy updates.

**Strengths:**

(1) it identifies and systematically addresses under-explored challenges at the intersection of quantization and RL, such as the divergence between behavior and proximal policies. The proposed ACR mechanism is both intuitive and effective, dynamically adjusting clipping bounds to sustain training stability over extended horizons.
(2)UAQ cleverly leverages invariant scaling to amplify weight updates relative to quantization noise, enabling the quantized model to reflect training dynamics more faithfully.
(3) the experimental validation is extensive, covering diverse model sizes (7B to 32B), tasks, and quantization formats (INT8/FP8), with consistent improvements over strong baselines like FlashRL.

**Weaknesses:**

(1) he UAQ method, though effective, introduces a hyperparameter (scale factor ss) that requires tuning, adding complexity. Also the adaption may add more computational overhead
(2)while QuRL is positioned between PTQ and QAT, its dependence on one-time scaling may limit adaptability to highly dynamic training regimes.

**Questions:**

(1) What are the implications of quantization on RL exploration-exploitation trade-offs, especially in tasks requiring high creativity or diversity?
(2) How would QuRL perform under more aggressive quantization (e.g., 4-bit), and what additional adaptations would be necessary?

---

> ### Author Response · Authors · 2025-11-19
> **Rebuttal by Authors**
>
> Thank you for your positive feedback. Please check our reply to your concerns/questions.
>
>
> > Q1.  *The UAQ method, though effective, introduces a hyperparameter (scale factor ss) that requires tuning, adding complexity. Also the adaption may add more computational overhead.*
>
> Ans: Thanks for your comment. The core issue UAQ addresses, the mismatch between weight updates (∝ αG ≈ 10⁻⁷ to 10⁻⁶) and quantization errors (≈ 10⁻⁵ to 10⁻⁴), is consistent across RL training due to trust region constraints. UAQ's scaling provides an **s² improvement** in the update-to-noise ratio (Eq. 12), while simply increasing learning rate α only provides linear improvement. As shown in Table 4, increasing learning rate (α = 1.5×10⁻⁶ or 2×10⁻⁶) actually *degrades* performance (29.06% and 26.66% vs. 31.25% with s=1.5), because larger learning rates violate trust region constraints and increase clipping, whereas UAQ amplifies updates *within* the existing optimization framework. Moreover, the scale we found is consistent; setting the scale to 1.5 works well for both DAPO and GRPO tasks.
>
> Regarding the implementation complexity, UAQ does not add any computational overhead during training. Before training, we modify the weight values in-place by multiplying the weight by s and dividing the layer norm weight by $s$. This preprocessing takes less than 1 minute to finish. No additional computation during training—the forward/backward passes, quantization operations, and RL updates all proceed identically to standard training.
>
> > Q2. *While QuRL is positioned between PTQ and QAT, its dependence on one-time scaling may limit adaptability to highly dynamic training regimes.*
>
> Ans: We agree with the reviewer that one-time scaling may limit adaptability to high dynamic training regimes. Under such circumstances, we can perform multiple times of scaling during the training if required. However, we’d like to point out **RL training is inherently low-dynamic due to trust region constraints:** Unlike supervised fine-tuning or pre-training where learning rates and update magnitudes can vary significantly across phases, RL training with PPO/GRPO enforces **trust region constraints** that keep weight updates small and relatively consistent throughout training, as shown in Appendix A (Fig. 9).  This stability is by design—RL algorithms explicitly constrain policy changes (via KL penalties, clipping, etc.) to prevent training instability.
>
> We view one-time scaling not as a limitation but as an **appropriate match to RL's constrained optimization dynamics**, while remaining extensible to more dynamic scenarios if needed.
>
> >Q3.  *What are the implications of quantization on RL exploration-exploitation trade-offs, especially in tasks requiring high creativity or diversity?*
>
> Ans: This is a very good question! During our experiments, we empirically found that quantization usually led to better convergence in the beginning of the training, but got stuck in the later stage of the training. Our implication is that quantization, though being used to accelerate the rollout, can be viewed as a random weight perturbation technique. This weight perturbation encourages the exploration of the actor by modifying the parameters, which is why it always has better convergence at the beginning. However, quantization may decrease the exploitation degree due to its lower precision numerics. There is a referenced paper that studies how random weight perturbation can help the exploration of traditional RL regimes.
>
>
>
>  > Q4. *How would QuRL perform under more aggressive quantization (e.g., 4-bit), and what additional adaptations would be necessary?*
>
> Ans: Thanks for your question. Under 4-bit quantization, the mismatch between the proximal actor and behaviour actor, and the mismatch between the weight update and the weight quantization will become larger. Although our techniques can mitigate them, they can accumulate for longer training iterations, and we expect to use mixed-precision adaptations for longer training iterations.
>
> To give an idea of 4-bit quantization, we run the GSM8K PPO with INT4 weight quantization experiments. The results are shown below.
>
> | Method  | Bitwidth | Accuracy |
> |---------|----------|----------|
> | RL      | BF16     | 55.35    |
> | RL      | INT4     | 0.00     |
> | FlashRL | INT4     | 47.44    |
> | QuRL    | INT4     | 52.70    |
>
> ### Reference
>
> Plappert, Matthias, et al. "Parameter space noise for exploration." ICLR 2018

---

### Official Review · Reviewer_PhnW · 2025-10-31

**Soundness:** 3
**Presentation:** 3
**Contribution:** 3
**Rating:** 4
**Confidence:** 3

**Summary:**

This paper introduces Quantized Reinforcement Learning (QuRL), a method to accelerate the rollout phase in reinforcement learning (RL) for large language models (LLMs) by quantizing the actor model during inference. The authors identify two key challenges: (1) training instability due to policy divergence between the full-precision and quantized actors, and (2) the mismatch between the magnitude of weight updates and quantization errors, which hinders the quantized model from capturing training dynamics. To address these, QuRL proposes Adaptive Clipping Range (ACR) to dynamically adjust trust region bounds based on policy divergence, and Update-Aware Quantization (UAQ) to amplify weight updates relative to quantization noise using invariant scaling. Experiments on GSM8K, AIME 2024, and DeepScaleR benchmarks demonstrate that QuRL achieves 20–80% faster rollout with INT8/FP8 quantization while maintaining competitive accuracy compared to full-precision baselines and outperforming naive quantization approaches.

**Strengths:**

1. The paper tackles a critical bottleneck in RL training for LLMs -- rollout latency -- by integrating quantization into the RL loop. The idea of decoupling quantized rollout from full-precision updates is innovative and addresses a timely problem in scalable RL.

2. The proposed solutions (ACR and UAQ) are well-motivated by empirical analyses of training instability and weight update mismatches. The ablation studies validate the contributions of each component.

3. Experiments span multiple RL algorithms (PPO, DAPO, GRPO), model sizes (7B–32B), and reasoning benchmarks, demonstrating broad applicability. Throughput gains (20–80%) are substantial and empirically validated.

4. The paper is well-structured, with clear explanations of challenges and methodologies. Figures effectively illustrate key issues like training collapse and weight update mismatches.

**Weaknesses:**

1. The work primarily contrasts QuRL with full-precision RL and naive quantization. Including comparisons to other efficiency methods (e.g., pruning, distillation) could better contextualize QuRL’s advantages.

2. Although ACR mitigates collapse up to 1,200 steps, the scalability to even longer training horizons (e.g., thousands of steps) is not tested. Stability under extended training remains an open question.

3. Robustness: How sensitive are ACR and UAQ to hyperparameters like the clipping bound or scale factor ? Are there guidelines for tuning them on new tasks?

**Questions:**

See weakness

---

> ### Author Response · Authors · 2025-11-19
> **Rebuttal by Authors**
>
> Thank you for the constructive feedback. Please check our reply to your concerns below.
>
> > Q1. *The work primarily contrasts QuRL with full-precision RL and naive quantization. Including comparisons to other efficiency methods (e.g., pruning, distillation) could better contextualize QuRL’s advantages.*
>
> Thank you for this insightful question. We want to clarify that quantization is a (probably the most) suitable efficiency method for RL rollout acceleration, as quantization is easy and cheap to apply, does not change model architecture, and is compatible with hardware.
>
> For pruning, after each RL update $\theta_{old} \leftarrow \theta$, the pruning mask would need to be recomputed or reapplied. Static masks become stale as the model evolves during RL training, while dynamic re-pruning and retraining after each RL step would negate efficiency gains.
>
> For distillation, maintaining a separate student model requires periodic re-distillation from the teacher (full-precision actor) to keep them synchronized. This re-distillation process itself requires forward/backward passes and is computationally expensive - potentially offsetting rollout savings.
>
> In summary, we believe that exploring quantization for RL rollout itself is a meaningful topic. In addition, quantization (as a numeric bitwidth approach) is orthogonal to the aforementioned other efficiency methods.
>
> > Q2. *Although ACR mitigates collapse up to 1,200 steps, the scalability to even longer training horizons (e.g., thousands of steps) is not tested. Stability under extended training remains an open question.*
>
> Ans: Thank you for this suggestion. We provide both empirical and theoretical evidence for long-horizon stability: **1. Our experiments demonstrated extended training:**  Our DeepScaleR experiments (Table 3) run for 1,600 total steps across three stages with increasing context lengths. QuRL maintains stable training throughout while baselines crash before completion (see our Appendix B for training dynamics). Furthermore, we continue our training of DeepScaleR with DeepCoder-1.5B datasets for an additional 300 iterations, resulting in nearly 2000 steps of total training. See our responses to Reviewer FcNP for results comparison.
>
> **2. Longer horizons require substantially more resources:** Testing 3,000+ step runs would require significantly larger computational budgets and typically involves larger models where RL training becomes even more expensive. As shown in ProRL (Liu et al., 2025), running RL in extended horizons requires 4x8 H100 GPUs for 16k GPU hours, which is beyond our current computational budget.
>
> **3. ACR provides theoretical stability guarantees:** ACR is designed to bound divergence through adaptive clipping. The key mechanism is that as quantization policies diverge ($r_{i,t}$ decreases), the original TIS mechanism will have a much tighter upper bound. Whereas our ACR increased this range dynamically and created a negative feedback loop that suppresses the tight clipping range. This self-correcting property, combined with the truncation bound C that caps maximum importance weights, should maintain stability beyond our tested horizons. Empirically, $KL(π_{behav} || π_{prox})$ stabilizes around 0.01 in Fig. 3a without exponential growth trends. We believe these factors provide substantial evidence for scalability, though we acknowledge that extremely long runs remain valuable future work.
>
> > Q3. *Robustness: How sensitive are ACR and UAQ to hyperparameters like the clipping bound or scale factor ? Are there guidelines for tuning them on new tasks?*
>
> Ans: Thanks for your question. We would like to point out that ACR does not introduce additional hyperparameters. Instead, it tries to correct the threshold given by a determined upper bound of the proximal-to-behavioral ratio. The clipping threshold is dynamically adjusted based on the quantized actor probability and the full precision old actor probability of that batch.
>
> As for the UAQ, the hyper-parameter scale is introduced to reduce the gap between weight quantization and weight update magnitude. We have conducted our ablation study of the scale versus learning rate choices in Section 5.3. Tuning scales is more robust than adjusting the learning rate since the scale factor brings $s^2\times$ smaller gap, while the learning rate only brings $s\times$ smaller gap.
>
>
> ### Reference
>
> Liu, Mingjie, et al. "Prorl: Prolonged reinforcement learning expands reasoning boundaries in large language models." NeurIPS 2025.

---

### Official Review · Reviewer_FcNP · 2025-11-02

**Soundness:** 3
**Presentation:** 3
**Contribution:** 3
**Rating:** 6
**Confidence:** 4

**Summary:**

This paper presents QuRL, a practical recipe to accelerate reinforcement learning for reasoning LLMs by performing rollouts with a low-precision actor (INT8/FP8) while keeping full-precision weights for policy updates, which effectively decouples the behavior (quantized) and proximal (full-precision) policies and necessitates careful importance sampling and trust-region control. They propose Adaptive Clipping Range (ACR), which absorbs the truncation factor into PPO’s clipping term, shrinking the effective range (with a fixed upper cap) to stabilize gradients and prevent late-stage collapse. The paper also introduces Update-Aware Quantization (UAQ), a one-time invariant scaling of transformer linear layers that simultaneously reduces quantization error and amplifies effective updates, yielding an s² improvement in update-to-noise ratio. On GSM8K with PPO, QuRL closes much of the gap between naive INT8 and BF16 (53.55% vs 55.35%) and also improves FP8 robustness relative to baselines. Overall, QuRL stabilizes decoupled PPO (via ACR) and makes low-precision rollouts learnable (via UAQ), offering a simple, engine-compatible path to faster RL for reasoning LLMs.

**Strengths:**

1. This paper studies an important problem of model quantization in reinforcement learning, which can help speed up the RL training process. The work decouples rollout and optimization by quantizing the actor for sampling while keeping full-precision weights for updates; the paper cleanly frames this as decoupled/off-policy PPO and motivates the need for trust-region/IS control.

2. Adaptive Clipping Range stabilizes late-stage collapse by adapting the clipping bounds to proximal-to-behavior ratios; Update-Aware Quantization (invariant scaling) reduces quantization error and amplifies effective updates with a simple, engine-compatible change.

3. The paper conducts a fair comparison to demonstrate the effectiveness of QuRL for RL training.

**Weaknesses:**

1. Marginal accuracy improvements in places: on GSM8K the final gap vs BF16 remains (~53.55 vs 55.35), and DeepScaleR’s reported average gain is modest (+1.7%), which may feel incremental relative to the engineering complexity.

2. Scale of core effectiveness studies: the detailed PPO accuracy study uses a 0.5B base model (Qwen2.5-0.5B); while throughput is shown for 7B–32B, end-to-end accuracy results beyond small models are limited. Therefore, the authors are suggested to conduct experiments on larger models and even MoE models to demonstrate the effectiveness of their method.

3. Nowadays more models focus on reasoning or agentic capabilities, which requires stronger long-context capabilities of model. The paper does not provide direct evaluations on thinking models or agentic models to study whether the proposed methods works under these settings.

**Questions:**

See weakness

---

> ### Author Response · Authors · 2025-11-19
> **Rebuttal by Authors**
>
> Thank you for your time to review our work and give insightful comments. Please check our detailed reply to your questions/comments.
>
> > Q1. “*Marginal accuracy improvements in places: on GSM8K the final gap vs BF16 remains (~53.55 vs 55.35), and DeepScaleR’s reported average gain is modest (+1.7%), which may feel incremental relative to the engineering complexity.*”
>
> We’d like to clarify that our primary contribution is RL training acceleration (20~80% faster) rather than improving baseline RL performance. The GSM8K results demonstrate that our method is capable of getting the same accuracy level as full-precision training under INT8/FP8 quantization.
>
> For the DeepScaleR results, while we reported the best accuracy during training, the main benefit of QuRL is training stability.  **FlashRL and vanilla INT8 RL will lose accuracy towards the end of the training** due to accumulated divergence between quantized and full-precision actors (see new Appendix B for training logs).  This demonstrates that ACR and UAQ solved an important stability problem that prevents existing methods from completing long-horizon RL training.
>
> On "engineering complexity": QuRL can be easily integrated into the existing RL frameworks like VeRL by modifying several lines of code in the loss function and a one-time scaling before training that can be done within one minute.
>
> > Q2. “Scale of core effectiveness studies: the detailed PPO accuracy study uses a 0.5B base model (Qwen2.5-0.5B); while throughput is shown for 7B–32B, end-to-end accuracy results beyond small models are limited. Therefore, the authors are suggested to conduct experiments on larger models and even MoE models to demonstrate the effectiveness of their method.”
>
> We want to clarify that our DAPO experiments were conducted on a 7B model (33.27 QuRL FP8 vs 31.67 RL BF16 on AIME 2024), much larger than 0.5B. We acknowledge that even larger-scale models (32B) RL training experiments would further validate our claims. Unfortunately, running full RL training on 32B models requires 4×8 H100 nodes, which is beyond our current resource budget. However, we emphasize that **smaller models are harder to be quantized:** As demonstrated by Liu et al. (2025) on the same DeepSeek-Distilled-Qwen family, the 32B model shows only ~6% accuracy degradation under 4-bit quantization, while the 1.5B model shows ~45% degradation. **Since QuRL successfully handles the more challenging smaller models (0.5B-7B), this suggests a strong transferability to larger scales** can be expected.
>
>
>
> > Q3. Nowadays more models focus on reasoning or agentic capabilities, which requires stronger long-context capabilities of model. The paper does not provide direct evaluations on thinking models or agentic models to study whether the proposed methods works under these settings.
>
> Thanks for your question. We’d like to clarify that our DeepScaleR experiments, which adopt DeepSeek-Distilled models, do focus on the CoT thinking capabilities. The average number of decoded tokens is 6000 per rollout output, and has a \<think\> token to do CoT. For agentic tasks, here we run the DeepCoder-1.5B task on top of the DeepScaleR experiments for an additional 300 steps. The evaluation of LiveCodeBench results is shown below, demonstrating our method’s scalability for agentic tasks.
>
> | Model    | Bitwidth | LCB  |
> |----------|----------|------|
> | RL       | BF16     | 23.5 |
> | RL       | INT8     | 20.2 |
> | QuRL+UAQ | INT8     | 22.9 |
>
>
> Given the limited time allocated for rebuttal, we are unable to perform more agentic tasks such as tool calling and computer use. However, the RL algorithmic designs should be similar to those of coding and math tasks. For RL training with extremely long contexts, this falls somewhat outside our scope and resource budget, but we still consider it a promising research direction to explore.
>
> ### Reference
>
> Liu et al., Quantization Hurts Reasoning? An Empirical Study on Quantized Reasoning Models, arXiv 2025.
>
> Luo et al., DeepCoder: A Fully Open-Source 14B Coder at O3-mini Level, Notion Blog, 2025.

---

### Comment · Area_Chair_zQy5 · 2025-11-27
**Follow-up on Rebuttal**

Hi,

The authors have submitted their rebuttal for paper. Please check the responses and confirm whether your questions are resolved.

Best,
AC

---

### Author Response · Authors · 2025-12-01
**Reviewer-Author Discussion Summary**

Dear Chairs and Reviewers,

### Thank you all for your time and valuable feedback on our work. Below, we summarize the main strengths and weaknesses identified, along with our response.

-----

## Strengths
1. **The investigated problem is important in the field.** We are the first to study efficient RL by efficient rollout inference with quantization. {FcNP, PhnW, xNJr}
2. **The experimental verification is thorough.** We test multiple RL algorithms, including PPO, GRPO, DAPO, and across different model sizes. {PhnW, xNJr}
3. **The proposed methodology**, Adaptive Clipping Range and Update-Aware Quantization, **effectively reduces the gap** between rollout inference and actor update. {FcNP, PhnW, xNJr}

------

## Weakness

1. Concerns about Larger Models.

**Response**: We conducted our DAPO experiment with 7B scale. Unfortunately, running full RL training on 32B models requires 4×8 H100 nodes, which is beyond our budget. But we argue that **smaller models are much less quantization-resilient, which can already validate QuRL’s effectiveness.**


2. Concerns about agentic tasks and the long training horizon over 1200 steps.

**Response**: In our rebuttal, we continued training our DeepScaleR experiment, and it now contains 2000 RL steps, including both thinking models and agentic tasks models. We also highlight that the baseline method would fail at these steps due to accumulated error.


3. Concerns about Hyperparameter sensitivity.

**Response**: Our method, ACR, does not add additional hyperparameters. It determines the bound based on the training hyperparameters. For UAQ, a scale parameter is introduced. We’ve conducted an ablation study on this and show it has better robustness.

-------

## Acknowledgements

We deeply appreciate all reviewers and the AC’s effort to review our work. Our initial rebuttal was posted on November 19th. Though not receiving any further feedback, we want to emphasize that Reviewer [PhnW] has increased his/her score to 6 before reverting. We hope our response has addressed all concerns and questions.

Best regards,



Authors.

---

### Meta-Review · Area_Chair_uHHT · 2026-01-07

**Summary:**

This work proposes the use of a quantized actor in order to speed up rollouts when post-training LLMs. The use of quantization introduces challenges which the authors address directly: the use of clipping to reduce the distance between the full-precision and quantized actor, and scaling weight updates to avoid negligible learning. The idea is novel and evaluation is robust, and all reviewers are generally positive about the work.

**Reviewer Concerns:**

All concerns were addressed by the authors. I list the most notable ones below.

## FcNP
- W1 (Marginal accuracy improvements): adequately addressed by authors, who also clarified that the "primary contribution is RL training acceleration (20~80% faster) rather than improving baseline RL performance"
- W2 (Scale of core effectiveness studies): The authors had run on larger models than the reviewer believed, and further clarified that "smaller models are harder to be quantized".

## PhnW
- W1 (work primarily contrasts QuRL with full-precision RL and naive quantization). This concern was well addressed by the authors, ending with "quantization (as a numeric bitwidth approach) is orthogonal to the aforementioned other efficiency methods", which is correct.
- W2 (Stability under extended training remains an open question). The authors provide a decent response to this. Although they were not able to train for as many steps as the reviewer was asking for due to computational constraints, they do provide reasonable empirical and theoretical justification for stability under extended training.
- W3 (Robustness). Well-addressed by the authors, as there are no new hparams introduced, aside from UAQ, for which they did provide ablation studies.

## xNJr
- W1 (tuning of UAQ hparam). The authors do provide a reasonable response to this and did conduct an ablation study of this hparam.
- W2 (QuRL dependence on one-time scaling may limit adaptability to highly dynamic training regimes). The authors provide a reasonable approach to mitigate this, but note the important point that "RL training is inherently low-dynamic due to trust region
constraints".
- Q1 (implications of quantization on exploration/exploitation). Reasonable response.
- Q2 (How would QuRL perform under more aggressive quantization?). The authors addressed this concern well by running extra experiments on 4-bit quantization.

**Reviewer Scores:**

- **FcNP:** Currently at 6, would have likely stayed at that score but slight possibility of an increase to 8.
- **PhnW:** Currently at 4, but they would have likely increased their score, given the author rebuttals.
- **xNJr:** Currently at 6, would have likely stayed at that score but slight possibility of an increase to 8.

---

### Decision · Program_Chairs · 2026-01-26

Accept (Poster)